# Targeting CD44 Receptor Pathways in Degenerative Joint Diseases: Involvement of Proteoglycan-4 (PRG4)

**DOI:** 10.3390/ph16101425

**Published:** 2023-10-08

**Authors:** Marwa M. Qadri

**Affiliations:** 1Department of Pharmacology and Toxicology, College of Pharmacy, Jazan University, Jazan 45142, Saudi Arabia; mqadri@jazanu.edu.sa; 2Inflammation Pharmacology and Drug Discovery Unit, Medical Research Center (MRC), Jazan University, Jazan 45142, Saudi Arabia

**Keywords:** CD44, PRG4, synovium, rheumatoid arthritis, osteoarthritis, gout, arthrofibrosis, therapeutic target

## Abstract

Rheumatoid arthritis (RA), osteoarthritis (OA), and gout are the most prevalent degenerative joint diseases (DJDs). The pathogenesis underlying joint disease in DJDs remains unclear. Considering the severe toxicities reported with anti-inflammatory and disease-modifying agents, there is a clear need to develop new treatments that are specific in their effect while not being associated with significant toxicities. A key feature in the development of joint disease is the overexpression of adhesion molecules, e.g., CD44. Expression of CD44 and its variants in the synovial tissues of patients with DJDs is strongly associated with cartilage damage and appears to be a predicting factor of synovial inflammation in DJDs. Targeting CD44 and its downstream signaling proteins has emerged as a promising therapeutic strategy. PRG4 is a mucinous glycoprotein that binds to the CD44 receptor and is physiologically involved in joint lubrication. PRG4-CD44 is a pivotal regulator of synovial lining cell hemostasis in the joint, where lack of PRG4 expression triggers chronic inflammation and fibrosis, driven by persistent activation of synovial cells. In view of the significance of CD44 in DJD pathogenesis and the potential biological role for PRG4, this review aims to summarize the involvement of PRG4-CD44 signaling in controlling synovitis, synovial hypertrophy, and tissue fibrosis in DJDs.

## 1. Introduction

Degenerative joint diseases (DJDs), classified as rheumatoid arthritis (RA), osteoarthritis (OA), and gout, are among the prominent causes of physical disability and impaired quality of life around the globe [1,2,3]. The prevalence and estimated disease burden of DJDs vary worldwide and are estimated to be higher in both economically developed countries and cities [1,4]. DJDs affect the whole joint, eventually leading to articular cartilage degeneration, synovial membrane inflammation, thickening of subchondral bone, osteophyte formation, and fibrosis [5,6,7,8,9,10]. Long-term, uncontrolled inflammation leads to destructive effects in the joint [8,10,11]. The etiopathogenesis of DJDs includes a complex network of risk factors involving predisposing genetic factors, environmental triggers, and mechanical and biochemical parameters that determine the predominance of destructive processes [1,12,13]. The pathogenesis of DJDs mainly involves the activation of innate immune cells and synovial fibroblasts in the synovial joint via the secretion of pro-inflammatory mediators, e.g., interleukin (IL-1β), IL-6, and tumor necrosis factor (TNF-α) [14,15,16]. Chronic activation of synovial lining cells results in synovial fibroblast proliferation, inflammation, and joint damage [8,11,14]. Adhesion molecules, such as cluster determinant 44 (CD44), are overexpressed as a feature of the inflammatory response in DJDs [17,18,19,20].

CD44 is involved in the regulation of a wide variety of cellular and physiological processes and is expressed in numerous cell types including immune system cells [21]. Overexpression of CD44 isoforms in the synovium has been associated with DJD development and progression [22,23,24,25,26]. Activation of CD44 by its ligands or joint microenvironments triggers the activation of many downstream pathways, e.g., mitogen-activated protein kinase (MAPK) and phosphoinositide 3-kinase/protein kinase B (PI3K/AKT), in order to mediate cell migration and proliferation [21,27]. Additionally, CD44 plays a major role in regulating inflammation and innate immunity via phagocytosis of cell debris [28,29,30,31], in vivo differentiation of monocytes into tissue macrophages [32], and tissue fibrosis [33,34,35,36]. Drugs commonly used to treat synovial inflammation in DJDs are inadequate in their therapeutic effect and have side effects that limit their use. CD44 has been considered a novel downstream regulator of pro-inflammatory and pro-fibrotic signals in DJDs. Proteoglycan-4 (PRG4) is a mucinous glycoprotein that plays a central role in mediating joint lubrication, synovial hemostasis, and suppressing inflammation [37,38,39]. PRG4 binds to the CD44 receptor [40] and exhibits anti-inflammatory, immunomodulatory, and antifibrotic effects in the joint. This review aims to summarize the contribution of PRG4-CD44 in the pathogenesis of DJDs.

## 2. Search Strategy and Selection Criteria

A search of PubMed from 2010 to the present was conducted and articles related to CD44-PRG4 signaling in rheumatoid arthritis, osteoarthritis, and gout were selected.

## 3. CD44 Receptor: Overview of Structure, Function, and Signaling

CD44 is a non-kinase, single-span transmembrane glycoprotein expressed in nearly all human tissues [41]. Many cellular processes, such as cell adhesion, proliferation, development, migration, phagocytosis, cytoskeleton rearrangement, and various inflammatory and metabolic diseases are regulated through CD44 signaling [42,43,44,45]. Previously, CD44 was described as a human erythrocyte surface antigen and lymphocyte homing receptor [46]. Currently, CD44 is being recognized as a glycoprotein adhesion molecule that mediates numerous signaling pathways in various types of cells [21]. In humans, the CD44 gene is encoded by 19 exons, and by 20 exons in mice [47]. Exons 1–5 encode the extracellular domain, while exons 16 and 17, and 18, 19, and 20 encode the transmembrane and intracellular cytoplasmic domains, respectively [44,47]. CD44 exists in standard (CD44s) and variant (CD44v) forms. Exons 16–20 encode the standard form, whereas exons 6–15 of the CD44v are produced through an alternative splicing [48]. CD44 isoforms share a common structure in which the N-terminal of the extracellular domain binds to several components, including hyaluronic acid (HA), chondroitin, fibronectin, osteopontin (OPN), and PRG4 [40,49], the membrane-spanning region, and the cytoplasmic domain, which transduces the signaling of several cytoskeleton-associated proteins and tyrosine kinases [49]. CD44s is predominately found on the membranes of smooth muscle cells, lymphocytes, and fibroblasts of most mammalian cells [50]. Expression of CD44v isoforms is mainly present in epithelial cells and keratinocytes [43]. The expression and distribution of CD44v isoforms have been associated with the progression and development of cancer, and autoimmune and inflammatory disorders [42,45,51]. 

Ligand-mediated CD44 signaling drives proliferation, migration, invasion, angiogenesis, cytoskeleton rearrangement, and inflammatory cytokine production [21,44]. CD44 activation triggers CD44–co-receptor complex formation in order to activate several intracellular signaling pathways. These co-receptors include c-mesenchymal-epithelial transition factor (c-Met), epidermal growth factor receptor [19], and transforming growth factor beta (TGF-β), among others [52]. CD44 promotes cytoskeleton change and activates vascular endothelial growth factors (VEGFRs), contributing to angiogenesis via ezrin/radixin/moesin (ERM) recruitment [53,54]. Furthermore, CD44 mediates cell invasion through the activation of Snail/β-catenin translocation and promotion of matrix metalloproteinase (MMP) expression [55,56]. CD44 stimulates cell proliferation via modulation of the Src/MAPK signaling pathway and enhances the activation of PI3K/AKT, leading to nuclear factor kappa B (NF-κB) translocation and inflammatory cytokine production [27,57,58].

CD44 plays a major role in the metabolism and cellular uptake of HA [59]. CD44 binds to several extracellular proteins, including osteopontin, collagen, MMPs, PRG4, and HA [60,61,62]. Moreover, the binding of CD44 to its antibody or HA results in protein phosphatase-2 A (PP2A) activation [63]. CD44 has been clearly associated with the development of a number of diseases, such as cancer, vascular diseases, tissue fibrosis, and degenerative joint diseases. Soluble CD44 protein, generated by alternative splicing or shedding from cell membranes by MMPs [64], has been found in body fluids such as cancer patient serum and arthritic synovial fluid and is associated with a poor disease prognosis [50,65]. CD44 and its signaling proteins have been discovered to be potential therapeutic targets. 

## 4. Proteoglycan-4 (PRG4): Localization, Structure, and Biological Function in the Joint

Lubricin/Proteoglycan-4 (PRG4) is a mucinous glycoprotein secreted from fibroblast-derived type B synoviocytes and chondrocytes [37,39]. PRG4 is encoded by a gene with approximately 345 kDa located at the surface of articular cartilage and fibroblast-like synoviocytes [66]. PRG4 is abundant (200–400 µg/mL) in synovial fluid, where it contributes to joint lubrication, reduces friction between cartilage surfaces, and prevents synovial overgrowth [37,38,67]. However, PRG4 levels in synovial fluid are reduced in patients with acute injuries [68,69]. Moreover, PRG4 gene expression is reduced in the synovium and cartilage of animal models of inflammatory arthritis [70,71]. Multiple reports have shown that synovial and cartilage PRG4 expression is reduced in rat, guinea pig, and mouse models of posttraumatic OA (PTOA) [68,70,72,73]. Synthesis of PRG4 and gene expression by chondrocytes are stimulated by TGF-β and reduced after exposure to IL-1β and TNF-α [74,75,76,77]. Camptodactyly-arthropathy-coxa vara-pericarditis (CACP) syndrome is an uncommon genetic illness that follows an autosomal recessive pattern of inheritance. This syndrome is caused by a mutation in the PRG4 gene and is characterized by the gradual enlargement of synovial tissue, the development of fibrillation, and the deterioration of articular cartilage. These manifestations can be observed in the PRG4 knockout (PRG4^−/−^) mice model [78,79]. In addition to lubrication, PRG4 regulates tissue homeostasis and modulates inflammatory and immune responses. Reports have shown that native human, recombinant form, and degraded PRG4 are able to reduce articular cartilage destruction and chondrocyte apoptosis, and enhance articular cartilage regeneration in inflammatory arthritis models [80,81,82,83,84]. 

Structurally, PRG4 is 1404 amino acids long with globular N- and C-termini and a central mucin domain [85]. The central mucin domain is highly glycosylated with O-linked β 1-3)Gal-GalNAc oligosaccharides. The glycosylation has an essential role in modulating PRG4’s biological effect. It is constructed to form a nanolayer that produces repulsive forces and imparts anti-adhesive effects [67,78,85,86,87,88]. PRG4’s N-terminus contains a somatostatin B-like (SMB) domain and heparin-binding site, whereas the C-terminus contains a hemopexin-like domain (Figure 1). PRG4 is analogous to vitronectin, as both contain hemopexin-like and SMB domains to regulate cell–extracellular matrix interactions [86]. Unlike vitronectin, PRG4’s central mucinous domain contains a repeating motif of KEPAPTT [38]. PRG4 from arthritic patients exerts increased sialylation, likely leading to altered glycosylation of the PRG4 [89]. PRG4’s C-terminus promotes the surface binding of the protein to different adhesion molecules [86]. Furthermore, PRG4 binds to L-selectin on the surface of mononuclear cells isolated from the synovia and synovial fluid of patients with RA in a glycosylation-dependent manner [90].

Biologically, PRG4 has an essential role in regulating synovial tissue macrophages in animal models. In the synovium of PRG4^−/−^, the expression of pro-inflammatory (M1; CD86+) macrophages is more predominant than that of anti-inflammatory (M2; CD206+) macrophages [91]. Interestingly, PRG4 restoration skews the total macrophages into anti-inflammatory phenotypes and reduces synovitis and synovial fibrosis [91,92]. Collectively, these investigations support the immunomodulatory effects of PRG4. 

PRG4 also competes with HA on its binding to CD44. The PRG4-CD44 signaling axis plays a crucial role in regulating the intricated interactions between resident and infiltrated immune cells in the synovium, with an impact on controlling the onset, progression, and resolution of inflammation and synovial tissue fibrosis.

## 5. Role of PRG4-CD44 in Regulating Synovitis and Synovial Hyperplasia in Rheumatoid Arthritis (RA)

Rheumatoid arthritis (RA) is well recognized as a highly severe autoimmune disorder and a prevalent type of chronic inflammatory arthritis, with a global incidence rate ranging from 0.5% to 1% of the population [2]. RA’s incidence is highest between the ages of 30 and 55 and the prevalence of this phenomenon is higher among women as compared to men [2]. RA affects the small joints of the hands and feet, fibrous and connective tissues, and is characterized by synovitis, synovial hyperplasia, and joint and cartilage destruction [93]. 

The exact etiopathogenesis of RA is still under investigation. A complex network of genetic, immunological, and environmental factors contributes to the development of RA. The presence of (HLA)–DRB1 is clinically linked to the development of the disease. According to reports, more than 80% of patients express the HLA-DRB1*04 allele, and people who have two of these alleles are more likely to develop RA [93]. Other alleles that are associated with the structural severity of RA include MHC, PADI4, MIF, and PTPN22 [93]. The pathogenesis of RA is very complex and mainly involves the activation of innate immune cells in a synovial joint [94]. 

The normal synovium comprises an intimal layer 1–2 cells thick and a synovial sublining layer. The intimal layer contains two types of cells, fibroblast-like synoviocytes and macrophages, that maintain synovial tissue homeostasis via the production of PRG4 and HA [95]. In contrast, the sublining layer includes blood and lymphatic vessels [95]. RA is initiated by the activation of autoantibodies and self-antigens, followed by a transition into a more inflammatory and pathological state [93]. Migrating lymphocytes and monocytes attack the synovium and degrade the articular cartilage via proteolytic cleavage of collagen and aggrecan [93]. Furthermore, the activated synovial macrophages and fibroblasts secrete several inflammatory mediators, such as cytokines, chemokines, growth factors, autoantibodies, and MMPs, in order to initiate the process of chronic inflammation [93,94]. The present clinical management of RA encompasses the utilization of nonbiologic therapeutics, such as non-steroidal anti-inflammatory drugs (NSAIDs), alongside biologic disease-modifying antirheumatic therapies, including methotrexate, immunosuppressants, and biological anti-inflammatory agents [93,94]. However, these agents are associated with extra-articular side effects and poor patient compliance. 

Previous investigations have underlined the role of CD44 activation in the RA synovium. The hallmark of synovitis in RA is overexpression of CD44 receptors. RA’s synovium contains a substantial amount of various CD44 isoforms, and patients with RA have synovial CD44 levels that are 3.5 times higher than those of OA patients [25]. The expression of distinct CD44 splice variants is strongly associated with RA severity [25]. J. Grisar et al. demonstrated that the expression of CD44v4, CD44v6, and CD44v7-8 in the RA synovia and endothelium positively contributes to fibroblast-like synoviocyte (FLS) proliferation and invasion activities [23]. The expression of CD44v3 on the surface of monocytes of RA patients is not found to be correlated with the development of RA, while CD44v6 expression on monocytes is significantly associated with the clinical disease activity index [96]. Furthermore, blocking CD44v4 in RA-FLS reduces FLS proliferation and IL-1ß mRNA expression in vitro [96]. 

One study reported that the proliferation of fibroblasts from RA patients depends on the expression of a unique CD44 variant (CD44v7/8) in vitro [97]. In addition, the synovial macrophages isolated from RA synovia contain a high amount of CD44v9 [97], suggesting the role of CD44 in the development and progression of RA. Injection of anti-CD44vRA monoclonal antibodies reduces synovitis in an animal model of collagen-induced arthritis as well as apoptosis of FLS isolated from RA patients [97]. Using a pharmacological agent that causes shedding of the extracellular surface of the CD44 receptor or modulates its activity is effective in animal arthritic models [29,98]. 

To study the significance of PRG4-CD44 in mitigating synovitis and synovial hyperplasia in RA, Al-Sharif et al. demonstrated that recombinant human PRG4 (rhPRG4) competes with HA in binding to CD44 in a concentration-dependent mechanism [40]. The study reported that PRG4^−/−^ mice show signs of joint degeneration and PRG4^−/−^ synoviocytes display a high level of CD44 expression compared to PRG4^+/+^ synoviocytes. Moreover, IL-1β and TNF-α treatment induces PRG4^−/−^ synoviocyte proliferation and rhPRG4 reduces cytokine-induced PRG4^−/−^ synoviocyte proliferation in a CD44-dependent mechanism [40]. The antiproliferative effect of rhPRG4-CD44 interaction is mediated by the ability of rhPRG4 to block NF-κB nuclear translocation, and this effect is reversed using the CD44 antibody [40].

## 6. Role of PRG4-CD44 in Regulating Synovial Inflammation and Arthrofibrosis in Osteoarthritis (OA)

Osteoarthritis (OA) is the most prevalent depleting degenerative musculoskeletal disease, affecting millions of people globally [3]. OA affects the articular cartilage and its components, including the synovium, ligament, capsule, and subchondral bone [99]. OA’s prevalence is increasing, and 78 million people will be affected in the United States by 2040 [3]. In OA, major weight-bearing joints such as the knee, hip, ankle–foot, and temporomandibular joint (TMJ) are affected [99]. The exact pathogenic mechanism of OA is still unclear. However, multiple risk factors have been linked to OA development, including genetics, aging, sex (female), obesity, hypertension, and joint injury [99].

OA is a heterogeneous disorder characterized by articular cartilage and ligament degeneration, joint space narrowing, osteophyte formation, synovial hypertrophy, and variable degrees of synovitis [100]. An intricate network of interactions between environmental, genetic, lifestyle, and biochemical factors, contributes to the development of the chronic inflammatory process [99,100].

Articular cartilage provides low-friction motion around the joints and is composed of an extracellular matrix, e.g., water, collagen, proteoglycan, and chondrocytes [101]. Chondrocytes maintain the average turnover of extracellular matrix constituents [101]. OA results from the failure of articular cartilage to maintain normal homeostasis. Numerous catabolic and anabolic biofactors influence matrix components’ synthesis rate and breakdown, including cytokines, growth factors, proteolytic enzymes, and joint trauma [100,101]. 

The synovium produces lubricin and hyaluronic acid to maintain synovial fluid hemostasis [95]. Synovitis is a common feature of OA development and progression. The activation of innate immune cells and complement is the main driving factor of OA synovial tissue inflammation and cartilage degeneration [102]. Synovial macrophages play a crucial role in driving destructive processes in OA. Synovitis is initiated by the activation of synoviocytes and macrophages via toll-like receptors (TLRs) by damage-associated molecular patterns (DAMPs) from articular cartilage, thus preserving the synovial membrane’s inflammation by synthesizing inflammatory mediators [100]. Osteoarthritic synoviocytes and chondrocytes produce prominent catabolic mediators such as matrix metalloproteinases, IL-1β, IL-6, and TNF-α, which mediate a state of chronic inflammation in OA [103,104]. These catabolic mediators promote circulating lymphocyte and monocyte infiltration into synovial tissue, resulting in cartilage destruction, synovial hypertrophy, and bone osteophytes [103,104]. OA is a multifactorial disease that involves the entire joint, and the pathological symptoms vary from one patient to another. There are no disease-modifying agents currently available for OA. Conventional management includes NSAIDs, intra-articular corticosteroids or hyaluronic acid injections, and physical therapy [105], considered the primary management of OA symptoms until total joint replacement is needed.

CD44 expression is a significant predictor of OA progression. Previous reports have indicated that the presence of CD44 in the synovium of OA patients can be correlated with disease severity [22,26]. S. Fuchs et al. stated that the presence of CD44v5 and CD44v6 in the cartilage and synovial fluid of patients with OA is correlated to histopathological changes associated with OA [106]. In order to broaden the role of PRG4-CD44 in OA pathogenesis, Alquraini et al. demonstrated that p50 and p65 contents in PRG4^−/−^ synoviocytes are higher compared to PRG4^+/+^ synoviocytes, consistent with a higher level of CD44 expression in PRG4^−/−^ synoviocytes [107]. Additionally, rhPRG4 prevents NFκB p50 and p65 nuclear translocation and IL-1β-induced OA synoviocyte proliferation in a CD44-dependent manner [107]. Further, rhPRG4 inhibits IκBα degradation, compatible with its ability to inhibit NFκB p50 translocation [107]. Moreover, rhPRG4 reduces IL-1β-induced metalloprotease enzyme expression, including MMP1, MMP3, MMP9, and MMP13, in OA fibroblasts [107]. rhPRG4 prevents COX2, IL-8, and aggrecanase-2 expression in IL-1β-stimulated human OA fibroblasts [107]. The anti-inflammatory effect of rhPRG4 is CD44-dependent, as the anti-CD44 antibody abolishes the rhPRG4 effect on pro-inflammatory mediators.

Synovial macrophages play a pivotal role in OA pathogenesis. In innate immune response, CD44 is shown to modulate Fcγ receptors and β_2_ integrin-dependent macrophage phagocytosis [28]. Interestingly, CD44 inversely regulates TLR receptor activation [98]. Our group demonstrated that neutralization of CD44 receptors, CD44 receptor knockdown, or binding by its ligand HA reduces TLR2 stimulation-mediated IL-1β and TNF-α expression and secretion, and NFκB translocation in human and murine bone-marrow-derived macrophages (BMDMs) [98]. Further, the study reported that the inhibitory effect of CD44 neuralization or binding by its ligand is mediated by increasing the activity of the intracellular PP2A enzyme [98].

In addition to CD44’s role in synovial inflammation, CD44 plays a vital role in mediating tissue fibrosis pathogenesis. Acharya et al. reported that CD44 knockout skin fibroblasts exhibit an altered cytoskeleton compared to the wild-type [33]. Synovial fibrosis is due to excessive collagen deposition by synoviocytes and is a common feature in the synovia of patients with advanced OA that contributes to joint pain and stiffness [108,109]. In synovial fibroblasts, TGF-β1 induces fibrotic alterations that are characterized by cell proliferation and accumulation of collagen type I. Furthermore, TGF-β1 facilitates the differentiation of OA synoviocytes into a myofibroblast-like phenotype, an effector cell of fibrosis, as determined by alpha-smooth muscle actin (α-SMA) expression, stress fiber formation, enhanced migration, and production of extracellular matrix components [110]. Strikingly, our group identified that increasing cAMP levels using forskolin increases PRG4 levels in OA fibroblasts [110]. Furthermore, synovial PRG4 treatment reduces collagen type I, and the differentiation of fibroblasts into myofibroblasts following TGF-β1 treatment in OA fibroblasts [92]. rhPRG also reduces TGF-β1-induced cell migration in OA FLS and murine fibroblasts [110]. Moreover, cultured murine PRG4^−/−^ synoviocytes with a high level of CD44 expression exhibit markers of fibrosis, including α-SMA protein and collagen type I, and synovial PRG4 treatment reduces these markers in PRG4^−/−^ synoviocytes [92]. PRG4 re-expression reduces fibrotic marker contents in cultured synovial fibroblasts in vivo [92]. The PRG4-CD44 axis regulates synovial fibroblast–myofibroblast transition in vitro and PRG4’s antifibrotic biological role is potentially due to its binding to the CD44 receptor, as CD44 facilitates rhPRG4 internalization by OA FLS, and rhPRG4 reduces p-Smad3 contents in TGF-β reporter cells in a CD44-dependent manner [92]. The PRG4-CD44 axis exhibits an anti-inflammatory and antifibrotic role in OA.

## 7. Role of PRG4-CD44 in Mitigating Inflammatory Response in Gout Arthritis

Gout is a common form of prolonged destructive arthritic condition with a prevalence of up to 4% and an incidence of 3 per 1000 person-years [1]. Gout occurs as a result of a disturbance in uric acid metabolism leading to the accumulation of monosodium urate (MSU) crystals in the synovial joint capsule, causing an inflammatory response that manifests as intense pain, swelling, and reddening of the skin [111]. If left untreated, gout complications may include the development of tophi and, eventually, joint deformity [111]. Gout is linked to the development of many comorbidities that negatively impact the quality of life, such as obesity, kidney disease, diabetes, and hypertension [111]. Nonetheless, gout is an established risk factor for the development of chronic degenerative joint diseases, e.g., OA. This association between gout arthritis and OA has been observed in many clinical trials [112]. Current treatment modalities mainly focus on managing acute flares using NSAIDs, glucocorticoids, and biologic agents, and preventing further attacks using urate-lowering therapy [113].

The molecular pathogenesis of acute gouty attacks that result in chronic tophaceous inflammation is not fully understood. It is thought that MSU crystals develop in the pericellular matrix of cartilage, and these crystals trigger an innate immune response that results in synovitis and cartilage degeneration [114]. MSU crystals activate synovial macrophages mediated by TLRs and CD44 to release the proinflammatory cytokine IL-1β, which drives the acute inflammatory response [29,115]. IL-1β stimulates neutrophil and monocyte infiltration into the synovium, a pathological hallmark of an acute inflammatory attack [116]. MSU crystal uptake by resident macrophages and infiltrated mononuclear cells provokes reactive oxygen species (ROS) production and activates the NLRP3 inflammasome [29,117]. We previously identified that TLR2 and TLR4 mediate MSU crystal phagocytosis, NFκB p65 translocation, NLRP3 inflammasome assembly, and IL-1β production in human and murine macrophages [29,115]. Bousoik et al. showed that removal of the extracellular region of CD44 significantly reduces MSU crystal phagocytosis, and IL-1β and IL-8 production in human and murine macrophages [29]. Further, we observed that rhPRG4 treatment reduces MSU crystal uptake by human and murine peritoneal macrophages in a PRG4-CD44-dependent manner [115]. We also showed that rhPRG4 colocalizes with CD44 on PRG4^−/−^ peritoneal macrophages and reduces MSU crystal phagocytosis and IL-1β secretion [115]. BMDMs from CD44 knockout mice (CD44^−/−^ BMDMs) show a reduction in MSU crystal uptake, and IL-1β mRNA expression and secretion compared to BMDMs from CD44 wild-type mice (CD44^+/+^ BMDMs) [29]. Surprisingly, CD44^−/−^ BMDMs display low levels of IL-1β and nuclear NF-κB p65 compared to CD44^+/+^ BMDMs [29]. We previously showed that rhPRG4 treatment reduces IL-1β and NF-κB p65 nuclear contents through a PRG4-CD44-dependent mechanism in OA and RA synoviocytes [40,107].

CD44 neuralization or engagement by its ligand results in the intracellular activation of the PP2A enzyme [29,98]. The PP2A enzyme is a serine/threonine phosphatase that controls several cellular processes, including cancer apoptosis, and cell growth modulation and survival [118]. Additionally, Bousoik et al. demonstrated that the activation of the PP2A enzyme can be achieved through the ablation of the extracellular region of CD44 [29]. Recent reports have shown that activation of the PP2A enzyme results in anti-inflammatory activity in multiple sclerosis [119]. We recently found that PP2A activation reduces MSU crystal internalization, IL-1β production, and NLPR3 inflammasome expression by human macrophages [120]. ElSayed et al. reported that rhPRG4 treatment activates PP2A, and reduces MSU crystal phagocytosis, caspase-1 activity, and IL-1β production in human macrophages, gout, and normal human peripheral blood mononuclear cells (PBMCs), while okadaic acid, a PP2A inhibitor, reverses rhPRG4’s effect [117]. Surprisingly, PRG4 inhibits urate production, and IL-1β and xanthine oxidoreductase expression by BMDMs and an in vivo model of acute gout [121]. In summary, PRG4-CD44 is crucial in the regulation of the etiopathogenesis of gout.

## 8. Summary and Future Considerations

PRG4 binds to the CD44 receptor and plays an important role in joint lubrication. PGR4-CD44 interaction reduces IL-1β- and TNFα-induced proliferation and NF-κB nuclear translocation in fibroblasts isolated from patients with RA and OA. Moreover, PGR4-CD44 interaction is found to abolish cytokine-induced MMP1, MMP3, MMP9, MMP13, COX2, IL-8, IL-6, and aggrecanase-2 expression in OA fibroblasts. Macrophages play a crucial role in driving inflammation and articular damage in DJDs. Hyluronan-CD44 is shown to downregulate TLR2 activation in human and murine macrophages. PRG4 binds to CD44 and contributes to the negative regulation of TLR2 activation. CD44 also contributes to the development of arthrofibrosis. Increasing cAMP levels using forskolin increases PRG4 levels in OA synoviocytes and reduces fibrotic markers in OA synoviocytes. PRG4-CD44 interaction reduces stress fiber formation, α-SMA, COL1A1, ACTA2, and PLOD2 in TGF-β-stimulated OA and PRG4^−/−^ fibroblasts. Furthermore, PRG4-CD44 inhibits the phosphorylation of Smad3 in TGF-β reporter cells and OA fibroblasts. Additionally, PRG4-CD44 interactions inhibit MSU crystal phagocytosis, NFκB p65 translocation, NLRP3 inflammasome assembly, and IL-1β production in human and murine macrophages. PRG4-CD44 interactions also activate the PP2A enzyme, and reduce MSU crystal phagocytosis and caspase-1 activity in human macrophages. In summary, PRG4-CD44 is an essential regulator of fibroblasts like synoviocytes and macrophage homeostasis in the joint, where lack of PRG4 expression triggers chronic inflammation and fibrosis, driven by persistent activation of tissue-resident macrophages. Targeting CD44 may prevent immune-mediated cytokine activation and chronic joint inflammation, and PRG4 may have a potential therapeutic application in mitigating synovial acute and chronic inflammation and fibrosis in patients with DJDs. Therapeutically, overexpressing PRG4 in the articular joint, using a viral-mediated gene delivery approach, may prove beneficial in reducing synovitis, arthrofibrosis, and cartilage degeneration in DJDs [122,123]. The role of the PRG4-CD44 signaling circuit in ameliorating the pro-inflammatory and pro-fibrotic signaling in DJDs is schematized in Figure 2. Key findings that support the involvement of CD44 in DJD pathogenesis and the biological role of PRG4 as an anti-inflammatory and antifibrotic therapeutic in DJDs are presented in Table 1. Continued research in this field will provide additional insights into the role of PRG4-CD44 signaling in mitigating the pro-inflammatory and pro-fibrotic signaling pathways in DJDs.

## Figures and Tables

**Figure 1 pharmaceuticals-16-01425-f001:**
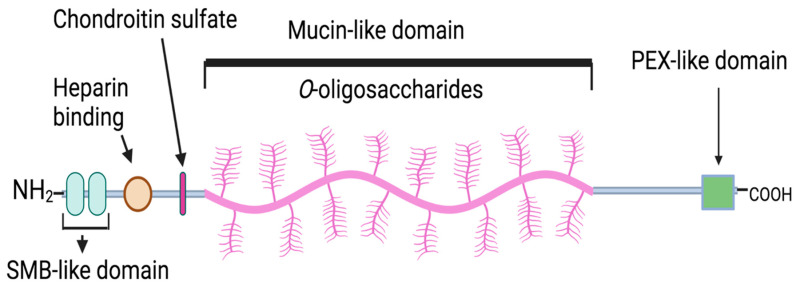
Schematic structure of full-length 1404 amino acid proteoglycan-4 (PRG4). PRG-4 contains a heavily glycosylated central mucin domain that imparts boundary-lubricating effects. PRG4’s *N*-terminus contains a somatostatin B-like (SMB) domain, heparin-binding site, and chondroitin sulfate. PRG4’s *C*-terminus contains a hemopexin-like domain. (Created with https://BioRender.com/).

**Figure 2 pharmaceuticals-16-01425-f002:**
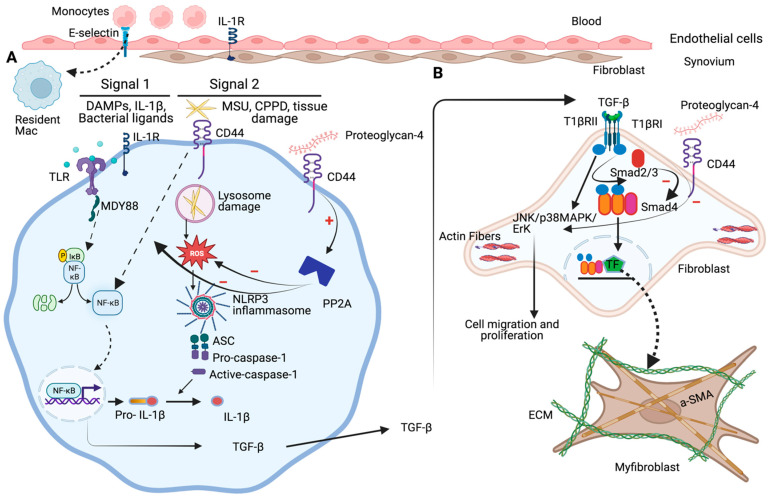
Role of the PRG4-CD44 axis in mitigating the pro-inflammatory and pro-fibrotic signaling pathways in DJDs. In response to injury, monocytes migrate into the synovial joint and differentiate into macrophages. Priming the resident macrophages by DAMPs, IL-1β, or bacterial ligands via TLRs or IL-1β receptor triggers signaling cascades that activate NFκB translocation, pro-IL-1β production, and NLRP3 inflammasome assembly, an innate cytosolic multiprotein complex that triggers IL-1β maturation. NFκB translocation induces transcription of proinflammatory cytokines, e.g., IL-1β and TNF-α, that drive the inflammatory response (**Signal 1**). Detection of endogenous danger signal, e.g., MSU crystals or extracellular matrix breakdown products by CD44 receptors (**Signal 2**), provokes ROS production, NLRP3 inflammasome assembly, and IL-1β secretion (**A**). Activated resident macrophages also produce profibrotic mediators, including TGF-β that directly activate resident fibroblasts and promote their differentiation into collagen-producing myofibroblasts that possess enhanced de novo α-SMA expression. TGF-β signaling pathway leads to Smad2/3 phosphorylation. The activated Smad2/3 form complexes with Smad4 and translocate to regulate downstream gene transcription. The TGF-β signaling pathway also activates the JNK/p38MAPK/Erk pathway, resulting in an increase in cell proliferation and migration (**B**). PGR4-CD44 interaction reduces endogenous danger internalization, DAMP-induced NF-κB nuclear translocation, IL-1β production, and NLRP3 inflammasome activation. The binding of PRG4 to CD44 receptor activates PP2A to inhibit ROS generation and NF-κB nuclear translocation. The PRG4-CD44 axis regulates synovial fibroblast to myofibroblast transition as exhibited by a reduction in Smad3 phosphorylation, α-SMA, collagen expression, fibroblast proliferation, and migration. (Created with https://BioRender.com/).

**Table 1 pharmaceuticals-16-01425-t001:** Role of PRG4-CD44 signaling in RA, OA, and gout pathogenesis.

Study	Model	Main Outcome
Al-Sharif et al. [40]	Binding assay using direct ELISA and surface plasmon resonance. Murine PRG4^−/−^ and PRG4^+/+^ synoviocytes immunostaining. IL-1β and TNF-α stimulated RA’s fibroblast.	rhPRG4 binds to CD44. PRG4^−/−^ synoviocytes display a high number of CD44 expression. rhPRG reduces cytokine-induced proliferation of RA’s fibroblast cells via inhibition of NF-κB nuclear translocation.
Alquraini et al. [107]	IL-1β stimulated osteoarthritic fibroblast-like synoviocytes.	rhPRG4 inhibits p50 and p65 nuclear translocation, IκBα degradation, and IL-1β-enhanced OA synoviocyte proliferation. rhPRG4 reduces IL-1β-induced metalloprotease enzyme expression, COX2, and aggrecanase-2 expression in OA fibroblasts.
Qadri et al. [98]	Human THP-1 and BMDMs stimulated with TLR2 ligand.	CD44 inversely regulates TLR receptor activation. Binding of the CD44 receptor to its antibody or ligand increases intracellular PP2A enzyme and reduces Pam3CSK4-induced cytokines expression and NFκB translocation in human and murine BMDMs.
Qadri et al. [92]	Internalization of rhPRG4 by OA FLS, TGF-β-stimulated OA FLS, and murine fibroblasts. Murine PRG4^−/−^ and PRG4^+/+^ synoviocyte immunostaining. TGF-β reporter cell line.	rhPRG treatment reduces cell migration and collagen type I, α-SMA, expression in TGF-β stimulated OA and murine fibroblasts in a CD44-dependent manner. Murine PRG4^−/−^ synoviocytes which had high levels of CD44 expression exhibit markers of fibrosis and rhPRG4 treatment reduced these markers in PRG4^−/−^ synoviocytes. rhPRG4 prevents p-Smad3 content in TGF-β reporter cells in CD44 dependent mechanism.
Qadri et al. [115]	PRG4^−/−^ peritoneal macrophages stimulated with MSU crystals.	rhPRG4 internalizes with CD44 in PRG4^−/−^ peritoneal macrophages. rhPRG4 treatment reduces urate crystals uptake and IL-1β secretion by peritoneal PRG4^−/−^ macrophages.
Bousoik et al. [29]	Incubation of anti-CD44 antibody with murine and human macrophages. Human THP-1, CD44^+/+^, CD44^−/−^ BMDMs stimulated with MSU crystals.	CD44 mediates urate crystals phagocytosis. Removal of the CD44 extracellular domain activates the PP2A enzyme and reduces MSU uptake, and IL-1β and NF-κB p65 nuclear levels in CD44^−/−^ BMDMs.
Qadri et al. and ElSayed et al. [117,120]	Human monocytes and gout and normal PBMCs treated with MSU crystals.	PP2A enhancement reduces urate crystal internalization and IL-1β secretion by human macrophages. rhPRG4 stimulates the PP2A enzyme and reduces urate crystal internalization, caspase-1 activity, and IL-1β levels in human macrophages, gout, and normal human PBMCs.
Elsaid et al. [121]	Urate production and xanthine oxidoreductase expression in granulocyte-macrophage colony-stimulating factor and IL-1β stimulated BMDMs.	rhPRG4 treatment reduces urate and xanthine oxidoreductase expression following granulocyte-macrophage colony-stimulating factor and IL-1β treatment in BMDMs.

## Data Availability

Data sharing is not applicable.

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
