# Peer review of "Targeting CD44 Receptor Pathways in Degenerative Joint Diseases: Involvement of Proteoglycan-4 (PRG4)"

_pharmaceuticals, 2023, doi:10.3390/ph16101425_

Round 1
Reviewer 1 Report
Targeting CD44 Receptor Pathways in Degenerative Joint Dis[1]eases: Involvement of Proteoglycan-4 (PRG4)
Abstract
The phrase "...processes associated with joint disease in DJDs remain unclear" seems somewhat ambiguous. Revise it.
It would be helpful to briefly state why the CD44 receptor is considered significant in DJD's pathogenesis, just for the clarity of readers unfamiliar with the topic.
The keywords are relevant and will assist in easy retrieval of the article. Consider adding "tissue fibrosis" and "therapeutic target" for a broader coverage.
Introduction
Kindly Check for typographical errors
Ensure all references are consistently formatted. They are presented as [1-3], [5-10], etc. If the intention is to denote a range, make sure all ranges are formatted this way.
"Upon reviewing the document, I noticed that there's a numbering discrepancy in the section headings. The section titled '1. CD44 Receptor: Overview of Structure, Function, and Signalling' is followed directly by '3. Proteoglycan-4 (PRG4): Localization, Structure, and Biological Function in the Joint.' It seems that section 2 is missing or not properly labelled. Kindly check and ensure that all sections are sequentially numbered and that no content is unintentionally omitted."
It's essential to address such issues, especially in scientific publications, to maintain clarity and consistency for readers.
The manuscript seems to lack a clear methodology section. Specifying your literature review process and the type of review (e.g., narrative, systematic) is crucial. Incorporating traditional review sections like Introduction, Methodology, Results, and Discussion will enhance the paper's structure and clarity. The manuscript, while rich in content, would benefit significantly from a more organized approach and presentation.
Author Response
Manuscript: Pharmaceuticals-2617798
Targeting CD44 Receptor Pathways in Degenerative Joint Diseases: Involvement of Proteoglycan-4 (PRG4)
Response to reviewer
The author would like to extend a sincere thank you to the reviewers for the time investment in critiquing the paper. The reviewers have provided very valuable comments that I have considered and addressed in the revised submission. Below, please find an overview of the changes that I have introduced. Following that is a detailed point-by-point response to reviewers. These changes are also highlighted in the submission.
Reviewer#1
Comment#1
Abstract
The phrase "...processes associated with joint disease in DJDs remain unclear" seems somewhat ambiguous. Revise it.
It would be helpful to briefly state why the CD44 receptor is considered significant in DJD's pathogenesis, just for the clarity of readers unfamiliar with the topic.
Response: The author has considered the reviewer’s recommendation
Action: Please refer to highlighted text for editorial changes (Page# 1, line 12) in the attachment below
Comment#2:
The keywords are relevant and will assist in easy retrieval of the article. Consider adding "tissue fibrosis" and "therapeutic target" for a broader coverage.
Response: The author has considered the reviewer’s recommendation
Action: Please refer to highlighted text for editorial changes (Page# 1, line 24) in the attachment below
Introduction
Comment#3:
Kindly Check for typographical errors
Response: The author has considered the reviewer’s recommendation
Action: Please refer Page# 1&2, line 28-61 in the attachment below
Comment#4:
Ensure all references are consistently formatted. They are presented as [1-3], [5-10], etc. If the intention is to denote a range, make sure all ranges are formatted this way.
Response: some of the references presented as hyphen between the reference numbers e.g.; [1-3] to denote a range and some of the references presented as comma between reference numbers to denote an individual reference number e.g.; [1,4]
Action: all references are consistently formatted, please refer to highlighted text for editorial changes (Page# 11-16, line 432-676) in the attachment below
Comment#5:
"Upon reviewing the document, I noticed that there's a numbering discrepancy in the section headings. The section titled '1. CD44 Receptor: Overview of Structure, Function, and Signalling' is followed directly by '3. Proteoglycan-4 (PRG4): Localization, Structure, and Biological Function in the Joint.' It seems that section 2 is missing or not properly labelled. Kindly check and ensure that all sections are sequentially numbered and that no content is unintentionally omitted."
It's essential to address such issues, especially in scientific publications, to maintain clarity and consistency for readers.
Response: All sections are sequentially numbered in the original uploaded word file. The section titled '1. CD44 Receptor: Overview of Structure, Function, and Signalling was number 2 in the original uploaded word file. The discrepancy in the section number is unintentionally omitted when the manuscript prepared in LaTex format.
Action: All sections are sequentially numbered, please refer to highlighted text for editorial changes (Page# 2, line 67) in the attachment below
Comment#6:
The manuscript seems to lack a clear methodology section. Specifying your literature review process and the type of review (e.g., narrative, systematic) is crucial. Incorporating traditional review sections like Introduction, Methodology, Results, and Discussion will enhance the paper's structure and clarity. The manuscript, while rich in content, would benefit significantly from a more organized approach and presentation.
Response: This is a narrative review and the author have considered the reviewer’s recommendation to include a searching strategy.
Action: Please refer to highlighted text for editorial changes (Page# 2, line 63-65) in the attachment below

Reviewer 2 Report
This review manuscript summarizes the how the CD44 plays an role in the development of degenerative joint diseases, and how the binding of proteoglycan-4 with CD44 can be a potentially valuable therapeutic target to address these diseases. The manuscript discusses Rheumatoid arthritis, osteoarthritis, and gout in further detail on how the PRG-4 and CD44 affect the pathological development of the disease. Overall the manuscript provides a clear summary on this specific topic. I recommend accept as is.
However, there are a couple of questions/suggestions: I think Figure 2 looks like it's made from BioRender, if so the website should be properly cited. The format of Section 7 is not consistent with rest of the manuscript. Lastly, I think it would make the manuscript better if the author can provide some insights on how to therapeutically utilize the benefit of PRG-4 in the degenerative joint diseases scenario.
Author Response
Manuscript: Pharmaceuticals-2617798
Targeting CD44 Receptor Pathways in Degenerative Joint Diseases: Involvement of Proteoglycan-4 (PRG4)
Response to reviewer
The author would like to extend a sincere thank you to the reviewers for the time investment in critiquing the paper. The reviewers have provided very valuable comments that I have considered and addressed in the revised submission. Below, please find an overview of the changes that I have introduced. Following that is a detailed point-by-point response to reviewers. These changes are also highlighted in the submission.
Reviewer#2
This review manuscript summarizes the how the CD44 plays an role in the development of degenerative joint diseases, and how the binding of proteoglycan-4 with CD44 can be a potentially valuable therapeutic target to address these diseases. The manuscript discusses Rheumatoid arthritis, osteoarthritis, and gout in further detail on how the PRG-4 and CD44 affect the pathological development of the disease. Overall the manuscript provides a clear summary on this specific topic. I recommend accept as is.
Comment#1
However, there are a couple of questions/suggestions: I think Figure 2 looks like it's made from BioRender, if so the website should be properly cited.
Response:
I thank the reviewer for the comment. Yes figures 1 and 2 were made using BioRender
Action: Please refer to highlighted text for editorial changes (Page# 4 line 151-152, and page#9 line 414) in the attachment below
Comment#2:
The format of Section 7 is not consistent with rest of the manuscript.
Response: The author has considered the reviewer’s recommendation
Action: Please refer to highlighted text for editorial changes (Page# 8, line 363-393) in the attachment below
Comment#3
Lastly, I think it would make the manuscript better if the author can provide some insights on how to therapeutically utilize the benefit of PRG-4 in the degenerative joint diseases scenario.
Response:
I appreciate reviewer’s comment. Many authors are affiliated with company and thus, without making an advertisement, it would be nice to speculate on potential therapeutic targets implicated by these studies.
Action: Please see highlighted text in the Summary and Future Consideration for editorial changes (Page# 8, line 385-387) in the attachment below

Round 2
Reviewer 1 Report
Dear Author,
Thank you for addressing the previous concerns and making corrections to your manuscript. Upon my latest review, the manuscript has been sufficiently improved. It is now closer to a publishable standard.
Before final acceptance, I kindly request you to thoroughly read it to ensure that all minor oversights, if any, are addressed. The essence of the work is clear and valuable; providing the highest level of polish will guarantee that it resonates well with the readership of this journal.
I look forward to seeing your work published.
Best regards,